# Antecedents of Self-Efficacy to Achieve Smoking-Behavior-Change Goals among Low-Income Parents Enrolled in an Evidence-Based Tobacco Intervention

**DOI:** 10.3390/ijerph192013573

**Published:** 2022-10-20

**Authors:** Mona L. Baishya, Bradley N. Collins, Stephen J. Lepore

**Affiliations:** Department of Social and Behavioral Sciences, College of Public Health, Temple University, 1301 Cecil B. Moore Ave, Philadelphia, PA 19122, USA

**Keywords:** smoking abstinence, self-efficacy, smokefree home, child tobacco smoke exposure

## Abstract

Previous studies have shown that greater self-efficacy (SE) to modify smoking behaviors during treatment improves long-term post-treatment outcomes. Little is known about factors that might enhance SE for smoking abstinence and for reducing children’s tobacco smoke exposure (TSE). The present study investigated hypothesized predictors of end-of-treatment SE to abstain from smoking and to protect children from TSE by conducting secondary multiple regression analyses of data (N = 327) from the Kids Safe and Smokefree (KiSS) behavioral intervention trial. KiSS aimed to reduce parental smoking and child TSE in urban, low-income, and minority communities. The results showed that partner support and initiating a planned quit attempt during treatment were positively related to SE to abstain from smoking and to reduce children’s TSE (all *p*’s < 0.001) at the end of treatment (EOT). Further, lower baseline nicotine dependence and the use of nicotine replacement were related to higher SE to abstain from smoking at EOT (*p* < 0.01), whereas more restrictive residential smoking rules and lower children’s TSE at baseline was associated with higher SE to reduce children’s TSE at EOT (all *p*’s < 0.05). These findings inform theory and future intervention design, identifying individual and social-environmental factors that might enhance smoking-behavior-change SE.

## 1. Introduction

Facilitating low-income parents’ smoking cessation and reducing their children’s tobacco smoke exposure (TSE) remain public health priorities due to the numerous harmful health consequences of smoking and children’s TSE [1,2,3,4,5]. Young children in low-income and African American households bear the greatest exposure risk and burden of TSE health consequences [2,3,6]. In addition, low-income and minority parents experience greater difficulty with cessation and children’s TSE reduction in evidence-based treatments than the general population of smokers [7,8]. These challenges are due in part to the broader contexts surrounding social norms that permit smoking, social and environmental deprivation, persistently elevated stress, and associated negative affect [9,10], which can undermine confidence in one’s ability to modify smoking behavior. Despite evidence that greater self-efficacy (SE) during smoking treatment improves subsequent treatment outcomes, little is known about factors that could potentially enhance SE for smoking abstinence and for reducing children’s TSE. The present study aims to examine antecedents of SE to abstain from smoking and to reduce children’s TSE among low-income parental smokers enrolled in evidence-based treatment.

SE describes one’s perceived ability to execute goal-oriented behavioral effort (e.g., eliminating smoking from the child’s bedroom) that is intended to lead to a specific outcome (e.g., child TSE reduction) [11]. It is a well-known behavioral mechanism of smoking-behavior change that predicts smoking abstinence [11,12]. For example, findings from prior research with low-income smokers in the Kids Safe and Smokefree (KiSS) behavioral intervention trial [13,14] suggested that higher SE at the end of treatment (EOT) predicted successful long-term, bioverified smoking outcomes. Specifically, the KiSS intervention was associated with improvements in SE at 3-month EOT (Time 2 [T2]), which, in turn, were associated with lower child cotinine (a biomarker of children’s TSE) and higher rates of parents’ smoking abstinence at 12-month (T3) follow-up [15,16].

In the present secondary analysis of KiSS data, we investigated factors at baseline and during treatment that could be linked to later SE at EOT to develop a deeper understanding of how we might improve SE in future treatments. Based on the social cognitive theory description of SE and principles of reinforcement, we hypothesized that greater effort and short-term success with smoking-behavior change, including initiating a quit attempt, greater practice with smoking-urge-management skills, and more days of abstinence during treatment would relate to both abstinence and TSE-reduction self-efficacy. We also hypothesized that SE would be bolstered by the more frequent practice of smoking-urge-management skills, which are goal-oriented behaviors that potentially can be improved through cognitive-behavioral therapy (CBT)-informed coping-skills training. Coupling goal setting with coping-skills training for urge management before a quit attempt may improve SE to abstain from smoking by providing smokers with opportunities to experience short-term control over their urges during periods of abstinence. Coping-skills training can include practicing compensatory coping strategies during urge-eliciting situations, or applying stimulus control strategies, during which smokers systematically reduce the number of permissible smoking contexts in their daily routine before quitting to create opportunities to practice urge-coping skills as they delay smoking [2,17]. A goal of pre-cessation stimulus control practice is to maintain progressively longer periods of short-term abstinence in newly defined non-smoking contexts. Such a goal-oriented effort has been shown to increase SE to abstain from smoking [12,18], which, in turn, can increase the likelihood of long-term abstinence and mitigate relapse risk [16,18,19,20,21,22,23].

Pharmacotherapy as well as coping-skills training to manage withdrawal symptoms after a quit attempt are standard elements in evidence-based smoking interventions. More severe withdrawal symptoms are associated with higher levels of nicotine dependence (a known barrier to cessation), which could inversely affect SE to abstain from smoking during intervention [24]. Thus, theoretically, the use of nicotine-replacement therapy (NRT) or medications prescribed for smoking cessation that mitigate nicotine withdrawal symptoms could increase SE to abstain from smoking. However, the degree to which NRT could facilitate parents’ SE and efforts to avoid smoking around their children and reduce children’s TSE is currently unknown. Exploring this potential association is warranted given that NRT could potentially improve one’s ability to avoid smoking around one’s children during a quit attempt.

Evidence-based smoking interventions also aim to enhance social support [25] and improve coping with psychological distress [26], factors that might influence smoking-behavior change via self-efficacy. Social cognitive theory maintains that self-efficacy is strongly influenced by social observations and principles of reinforcement during social interactions [11]. Interventionists and others in a smoker’s environment could foster a smoker’s confidence in smoking-behavior change with encouragement and emotional support, goal reminders and instrumental support, and positive reinforcement for short-term goal achievements. Emerging evidence supports the link between social support and self-efficacy to abstain from smoking [27,28], but to our knowledge, the link between social support and self-efficacy to reduce child TSE has not been explored. Other psychosocial foci in evidence-based interventions relate to well-established evidence of negative affect (e.g., stress, boredom, and depressive symptoms) as a key barrier to smoking cessation. The results from one study suggest that a higher level of depressive symptoms was related to lower SE to abstain from smoking [26]. Theoretically, like higher levels of nicotine dependence, higher levels of depressive symptoms increase the challenge to initiating and maintaining the effort to manage urges during a quit attempt, undermining opportunities for short-term goal-oriented successes that could foster SE.

Other less mutable factors, such as smokers’ age, could correlate with smoking-behavior-change self-efficacy. For example, compared to younger smokers, older smokers may have greater motivation to quit smoking due to the greater likelihood of experiencing health-related consequences of tobacco smoking. Established evidence points to associations between greater motivation to change and the improved likelihood of treatment success. This association could be influenced by SE given that more motivated clients may have greater intervention engagement, which, in turn, would increase the likelihood of experiencing positive reinforcement for goal-oriented effort and short-term successes during treatment. Additionally, older smokers have a longer history of prior quit attempts and, consequently, would have a larger repertoire of previously practiced strategies that could bolster SE prior to initiating subsequent quit attempts. However, little is about the potential association between smokers’ age and their self-efficacy to reduce children’s TSE. Thus, examining age-SE associations is warranted.

In this study, we investigate SE to abstain from smoking and SE to reduce child’s TSE in a sample of low-income smoking parents, a population for which understanding factors affecting SE for smoking-behavior change is particularly important. Using data from the KiSS trial, cognitive-behavioral theories and extant empirical evidence guided the identification of baseline demographic and psychosocial factors as well as measures of behaviors and social support during treatment that were hypothesized to predict subsequent SE outcomes at end of treatment (EOT). Specifically, to guide our analyses, we hypothesized that older parent age and greater social support, as well as measures capturing short-term effort and achievement with smoking-behavior change during treatment (e.g., initiating a quit attempt), would relate to both measures of self-efficacy at the EOT. We hypothesized that biological (i.e., lower baseline nicotine dependence level) and social (i.e., social support for cessation) factors, in combination with the behavioral factors described above, would enhance end-of-treatment SE to abstain from smoking. In contrast, we hypothesized that self-efficacy to reduce children’s TSE would be influenced more by psychosocial factors affecting home environments and child TSE (e.g., residential smoking restrictions and living with other smokers at baseline) as well as social support for TSE reduction.

## 2. Methods

This study examined secondary data from the multilevel KiSS intervention trial (detailed methods described previously) [29]. The trial used a parallel 2-group randomized controlled design with three assessment points at baseline (T1), 3-months end-of-treatment (T2), and 12-month follow-up (T3) [29]. All data in this analysis were self-reported by participants in computer-assisted structured telephone interviews conducted by research staff. Before study randomization, all participants received a pediatrician-initiated tobacco intervention in primary care based on pediatrics’ best practice guidelines, “Ask, Advise, Refer (AAR).” To guide a standardized AAR intervention within routine clinic encounters in partnering clinics, trial principal investigators modified electronic medical records and trained pediatric providers to follow electronic prompts to identify parental smokers (“Ask”); advise them about the dangers of child TSE and the benefits of smokefree homes and cessation treatment (“Advise”); and connect them to cessation treatment, including a referral to the trial (“Refer”). After T1 data collection, participants were randomized to either telephone-based behavioral counseling intervention consisting of skills training, support, and problem-solving, to promote child TSE reduction and parental cessation (AAR + counseling), or to a telephone-based attention control nutrition education intervention (AAR + control). The present study examined hypothesized predictors of subsequent T2 SE to abstain from smoking and SE to reduce children’s TSE measured using predictor variables from the T1 assessment (e.g., the nicotine-dependence level, whether the parent lived with other smokers in the home), as well as predictors reflecting behavior change effort prior to T2 during treatment (e.g., partner support, whether a quit day was achieved).

### 2.1. Participants

Participants were recruited from pediatrician clinics in three large urban pediatric systems that served low-income, racial/ethnic minority families (for details, see [29]).

### 2.2. Outcome Variables

SE to abstain from smoking was measured with the 12-item self-efficacy for avoiding smoking scale [30]. Participants rated how sure they were (1 = not at all to 4 = very) that they could avoid smoking in different situations that typically elicit smoking. Items were summed, with higher scores indicating higher self-efficacy to abstain from smoking. The reliability of the scale was good (Cronbach’s alpha = 0.89). We used a parallel 3-item measure to assess SE to reduce children’s TSE [30]. Participants rated how sure they were (1 = not at all to 4 = very) that they could create and maintain a smokefree home and keep their child away from other smokers’ tobacco. Items were summed, with higher scores indicating higher self-efficacy to abstain from smoking. The reliability of the scale was good (Cronbach’s alpha = 0.90).

### 2.3. Independent Variables

Independent variables included hypothesized predictors of SE outcomes. The baseline participant characteristics of interest included parent age, the number of other smokers at home, and depressive symptoms [31]. Depressive symptoms weremeasured with the short form of the Center for Epidemiological Studies Depression Scale, which is validated, reliable, and useful in smoking studies [31]. Baseline smoking and TSE variables included nicotine dependence [32], the average number of cigarettes smoked per day, residential smoking rules, and reported child TSE from all sources and locations. Nicotine dependence was measured using the reliable and validated Fagerstrom Test for Cigarette Dependence [33]. Variables related to behavioral achievement and perceived social support during the treatment period included whether participants successfully initiated a quit attempt (achieving at least 24 h of abstinence after a planned quit attempt), the number of days they maintained abstinence, the frequency with which they used urge coping strategies, whether they used NRT for at least two weeks, and how much support they received for their efforts with TSE reduction and smoking abstinence [34]. The treatment condition was used as a control variable.

### 2.4. Analytical Approach

Analyses were conducted in SAS software, version 9.4. First, the data were examined for normality and outliers. If there were extreme outliers, variables were dichotomized based on the median value, which was the case for the variable indicating total secondhand smoke exposure at baseline. Data from the KiSS trial was assumed to be missing at random [15,16]. Missing data were addressed by using the PROC MI function in SAS 9.4 to carry out multiple imputations [35]. Multiple regression analysis (MRA) was used to identify the unique predictors of the two SE outcomes. We reduced the number of variables included in the MRA by first conducting partial correlation analyses that controlled for treatment condition. Any variable that was significantly correlated (*p* < 0.05) with an SE outcome in the partial correlation analysis was included in the MRA. MRA used backward stepwise methods with an assessment of multicollinearity and R2 to determine the most parsimonious models with the most variance explained in predicting SE outcomes. An alpha level of 0.05 was used to determine statistical significance in the MRA.

## 3. Results

Detailed participant sample characteristics were described previously [14]. Participants (N = 327) were primarily female (83%) and Black/African American (83%), with 78.6% living below the poverty level. Parents smoked an average of 11.4 cigarettes per day, and 46.2% of the participants lived with at least one other smoker. Table 1 shows the partial correlations between the hypothesized factors and SE outcomes, controlling for treatment condition. Variables not included in subsequent MRA included parent education, the average cigarettes smoked per day, the number of residential smokers, and the level of depressive symptoms.

Table 2 shows the results of the MRA modeling of the two SE outcomes. Variables retained in the SE to reduce children’s TSE model with positive, independent, and significant associations with the outcome included participation in AAR + counseling, quit attempt initiated, the level of partner support, the presence of residential smoking rules, and parent age. The T1 level of child TSE from all sources was a negative and statistically significant predictor in the model. Variables retained in the SE to abstain from smoking model with statistically significant and positive, independent associations with the outcome included participation in AAR + counseling, quit attempt achieved, partner support, and NRT use. Nicotine dependence was a negatively associated and statistically significant predictor in the model.

## 4. Discussion

Increased short-term SE is a well-established predictor of long-term health behavior achievement, including smoking cessation outcomes. This secondary analysis improves the understanding of factors that may increase SE in smoking-behavior change among low-income parents, a population that is known to have greater challenges in evidence-based treatments focused on reducing child TSE and cessation. The results largely supported the hypotheses. For example, partner support during treatment was an independent predictor of SE in both models, whereas baseline (T1) nicotine dependence and NRT use during treatment were significant predictors in only the SE for abstinence model, and baseline residential smoking restrictions and level of child exposure were unique predictors in the SE for TSE reduction model.

Of the variables retained in both models, initiating a quit attempt during treatment accounted for the greatest variance in predicting both T2 SE outcomes compared to the other predictor variables. This evidence supports social cognitive theory [36] and broader established evidence that health behavior-related SE increases as a consequence of prior health behavioral achievements in interventions [11]. Interestingly, initiating abstinence on a planned quit day predicted SE, but SE was not influenced by either greater use of urge-coping skills or a longer period of abstinence during treatment. Initiating a quit attempt with at least one day of abstinence would offer the experience of some success in managing withdrawal along with positive reinforcement for the achievement. Such experience would likely be a more salient and SE-bolstering than the experience of practicing urge-coping skills in general (whether or not a quit attempt was made). However, theoretically, the longer an ex-smoker maintains effort to remain abstinent, the more their SE to maintain ongoing abstinence in the future should increase. Perhaps the null effect of longer abstinent days on SE suggests that maintaining abstinence over time, especially past the initial weeks following a successful quit attempt, is perceived as less of an achievement compared to the effort to initiate quitting and overcome nicotine withdrawal.

Enrollment in the AAR + counseling condition was another variable significantly associated with T2 SE in both models, consistent with previous analyses [15,16] and reflective of one of the KiSS intervention’s a priori counseling process goals to assist participants in goal setting and provide positive reinforcement for short-term goal-oriented effort—processes that theoretically should increase self-efficacy in smoking-behavior change. Perceived social support was also a significant positive predictor in both models. Supportive partners may provide encouragement and positive reinforcement for participants’ smoking-behavior change efforts, thereby increasing their intervention engagement and opportunities for short-term goal-oriented success, which could subsequently bolster SE. The potential for social support to enhance SE during smoking intervention highlights the importance of intervention elements that build social support in standard treatment.

Interestingly, both models retained NRT use as a predictor, but it was only a significant factor predicting SE to maintain abstinence, as hypothesized. NRT use could indeed bolster abstinence SE by facilitating greater control over smoking urges and nicotine withdrawal symptoms. However, regarding TSE-reduction self-efficacy, parents’ ability to reduce their children’s TSE extends beyond their control, and can be influenced by other factors within the larger socioenvironmental milieu in the home and community. Additionally, in the SE to maintain abstinence model, higher T1 baseline nicotine dependence was related to lower T2 SE to abstain from smoking. The association between dependence and SE is consistent with previous studies [24]. Greater substance dependence typically translates into greater difficulty in achieving and maintaining abstinence. Hence, the greater difficulty experienced trying to quit could undermine subsequent SE.

Parent age was retained in both SE models. As described earlier, perhaps older smokers have greater experience with prior smoking-behavior change and, hence, elevated SE compared to younger smokers. However, older age was only a significant independent predictor in the SE to reduce TSE model.

As predicted, only the SE to reduce child’s TSE model retained T1 child TSE and home-smoking policies as significant predictors of higher SE. It is understandable how parents who have already achieved greater TSE protection for their children (lower baseline child TSE) and have adopted more restrictive home-smoking rules would have greater confidence in their ability to reduce their children’s TSE through treatment. In parallel with the interpretation of the nicotine dependence–SE association, parents who are aware of higher levels of children’s TSE and have less restrictive rules around indoor smoking prior to initiating treatment may experience greater challenges in adopting more restrictive indoor smoking rules and protecting their children from TSE, hence undermining their SE to reduce TSE.

While the results of this study contribute to our understanding of factors influencing smoking-behavior-change SE, there were limitations to this study. For example, the data were limited in variables that evaluate change in SE over time. The original KiSS trial limited the inclusion/exclusion criteria that defined the range of ages, types of smokers, pregnant smokers, and the generalizability of the findings. Another limitation is the reliance on self-report, which is subjective. Nonetheless, this secondary analysis identified participant characteristics as well as factors representing social support and smoking-behavior change during treatment that predicted higher T2 self-efficacy—a cognitive factor that prior research showed predicted 12-month (T3) bioverified abstinence [16] and lower child cotinine [15]. Because of the influence of SE as an important mediator of the association between evidence-based behavioral smoking intervention and long-term smoking outcomes [37,38], the results of the present study inform important targets of intervention that can enhance SE and, in turn, improve the likelihood of long-term abstinence and child TSE reduction.

## 5. Conclusions

Low-income populations of tobacco smokers bear greater tobacco use- and exposure-related morbidity than the general population of smokers [39]. Future treatments focused on smoking abstinence could be improved to meet this challenge, with more potent social support elements that can compete with ongoing industry promotion and social norms in communities that minimize the importance and benefits of smoking abstinence, smoke-free homes, and children’s TSE reduction. Because of evidence pointing to the importance of SE in facilitating long-term smoking-behavior change, results from the present paper suggest that emphasizing and reinforcing even a short-term quit attempt during the treatment period can improve SE, perhaps through the experience of short-term goal-oriented successes that can be reinforced by counselors and significant others. A second critical takeaway is evidence that bolstering social support is important as it is related to both SE outcomes. Thus, the findings from this study can guide iterative improvements to future intervention strategies targeting low-income smokers.

## Figures and Tables

**Table 1 ijerph-19-13573-t001:** Means and correlations of variables related to self-efficacy outcomes at end of treatment (T2) (N = 327).

	Mean/N	(SD)/%	Partial Correlations Controlling for Condition			
			Self-Efficacy to Reduce Children’s TSE		Self-Efficacy to Abstain from Smoking	
*Demographics*						
Parent age T1	33.31	8.60	0.14	*	0.00	
Education T1						
Less than high school	89.00	27.22	0.05		−0.02	
High school or GED	238.00	72.78				
*T1 Smoking and exposure patterns*						
Average cigarettes smoked per day T1	11.45	7.56	0.03		−0.07	
Nicotine dependence level T1	4.07	1.92	−0.04		−0.16	**
Child TSE-all sources T1	0.50	0.50	−0.19	**	−0.03	
*Behavioral achievements during treatment*						
24 h quit day achieved before T2						
Yes	202.00	61.77	0.26	***	0.40	***
No	105.00	32.11				
Missing	20.00	6.12				
Abstinent days during treatment T2	13.46	21.85	0.25	***	0.46	***
Use urge coping during treatment T2	2.59	0.71	0.22	***	0.32	***
Used NRT at least two weeks T2						
Yes	71.00	21.71	0.14	*	0.17	**
No	256.00	78.29				
*Psychosocial*						
Partner support T2	25.65	9.40	0.39	***	0.28	***
Residential smoking rules T1	2.01	0.94	0.22	***	0.04	
Parents live with other smokers T1						
Yes	151.00	46.18	0.04		0.05	
No	176.00	53.82				
CES-D score T1	10.43	6.28	−0.08		−0.02	

T1 = baseline; T2 = End of Treatment; NRT = Nicotine Replacement Therapy; CES-D = Center for Epidemiologic Studies Depression scale. * *p* < 0.05, ** *p* < 0.01, *** *p* < 0.001.

**Table 2 ijerph-19-13573-t002:** Predictors of self-efficacy to abstain from smoking and self-efficacy to reduce children’s TSE (N = 327).

Predictor		Beta		SE
*Self-efficacy to maintain abstinence (R^2^ = 0.26)*				
	Quit day achieved before T2	7.60	***	1.09
	Treatment condition	2.63	*	1.02
	Using NRT at least two weeks	2.81	*	1.39
	Nicotine dependence T1 (FTND)	−0.85	**	0.27
	Partner support T2	0.22	***	0.06
	Parent age T1	0.01		0.06
*Self-efficacy to reduce children’s TSE (R^2^ = 0.28)*				
	Quit day achieved before T2	0.83	***	0.20
	Total exposure from all sources T1	−0.54	**	0.19
	Treatment condition	0.38	*	0.19
	Residential smoking rules T1	0.34	***	0.10
	Using NRT at least two weeks	0.34		0.25
	Partner support T2	0.06	***	0.01
	Parent age T1	0.03	*	0.01

T1 = baseline; T2 = End of Treatment; FTND = Fagerstrom Test of Nicotine Dependence; NRT = Nicotine Replacement Therapy. * *p* < 0.05, ** *p* < 0.01, *** *p*< 0.001.

## Data Availability

Data are not publicly available.

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
