# Peer review of "Antecedents of Self-Efficacy to Achieve Smoking-Behavior-Change Goals among Low-Income Parents Enrolled in an Evidence-Based Tobacco Intervention"

_ijerph, 2022, doi:10.3390/ijerph192013573_

Round 1

Reviewer 1 Report

Congratulations to the research team for this manuscript! It is of significance to the field, it addresses a gap in research and is presented very well, with impeccable style - it is very easy to read and comprehend, and it manages to convey the purpose of the study very well, along with its contribution to the current body of knowledge. The limitations of the study are well-presented, for a cross-sectional design with a relatively small sample. I will recommend that the manuscript be accepted in its current form - the first time I make such a recommendation!

Later Note: The authors of this manuscript, who are obviously seasoned researchers, do not know, for some reason that is foreign to me, that it is advisable, when sending an article in the first round, to also include some passages that the reviewers can criticize, so that you can have them afterwards removed, thank them for their good work in improving your manuscript, and happily go on with your lives (both reviews and authors). This recommendation extends beyond this manuscript, for which it is, unfortunately, too late to proceed this way. Someone very smart said this to me once – he is now Department Head.

Author Response

Thank you for such a kind, generous, and complimentary review. We appreciate all the time that you took to review our manuscript and were really honored to receive such a nice recommendations. 

Reviewer 2 Report

The authors performed secondary analysis of data of the Kids Safe and Smokefree behavior intervention trial and evaluated factors associated with parents’ self-efficacy (SE) to abstain from smoking and to reduce their children’s tobacco smoke exposure.  The analysis is straightforward, and the manuscript is well written. The findings that factors affecting parents’ PE for abstaining from smoke are different from those for reducing their children’s exposure to tobacco smoke in some points are important for future intervention.  However, the introduction is a little redundant, and similar description are found in the discussion.

Specific comments

Hypothesis and aim should be simple. The authors’ hypotheses for this study which were mentioned in L52-60 are similar to the hypothesis mentioned in L119-125.

Detailed description and discussion is not necessary for all the factors which might affect smokers’ PE. The authors can focus on several important points.  Description regarding smoking intervention and social support in L83-90 are similar to that in L244-251. Another description regarding parents’ age in 100-111 are similar to that in L265-272.

Author Response

IJERPH KiSS Self-Efficacy manuscript. Revision.

Thank you for the timely review of our original submission and the opportunity to revise and resubmit our manuscript titled, "Antecedents of self-efficacy to achieve smoking behavior change goals among low-income parents enrolled in an evidence-based tobacco intervention." Please find the authors’ responses to reviewer comments below in italics.

Editor’s Summary

The authors performed secondary analysis of data of the Kids Safe and Smokefree (KiSS) behavior intervention trial and evaluated factors associated with parents’ self-efficacy (SE) to abstain from smoking and to reduce their children’s tobacco smoke exposure (TSE).  The analysis is straightforward, and the manuscript is well written. The findings that factors affecting parents’ PE for abstaining from smoke are different from those for reducing their children’s exposure to tobacco smoke in some points are important for future intervention.  However, the introduction is a little redundant, and similar description are found in the discussion.

Thank you for this summary and comment. We carefully went through the introduction and discussion sections to reduce redundancy.

Specific reviewer comments

Hypothesis and aim should be simple. The authors’ hypotheses for this study which were mentioned in L52-60 are similar to the hypothesis mentioned in L119-125.

Thank you for this observation. Our response follows the summary of the reviewer’s comments about our hypotheses between lines 52-60 and 119-125.

Original L52-60: In the present secondary analysis of KiSS data, we investigated factors at baseline and during treatment that could be linked to later SE at EOT to develop a deeper understanding of how we might improve SE in future treatments. Based on the social cognitive theory description of SE and principles of reinforcement, we hypothesized that initiating a quit attempt and more days of abstinence during treatment would relate to greater SE to both maintain abstinence and reduce children’s TSE. We also hypothesized that SE would be bolstered by more frequent practice of smoking urge management skills, a goal-oriented behavior that potentially can be improved through cognitive-behavioral therapy (CBT)-informed coping skills training. Coupling goal setting with coping skills training.

We have now clarified this section to read, starting in line 55:  “… we hypothesized that greater effort and short-term success with smoking behavior change, including initiating a quit attempt, greater practice with smoking urge management skills, and more days of abstinence during treatment would relate to both abstinence and TSE-reduction self-efficacy.”

Originial L119-125: Specifically, we hypothesized that key demographic, social, and behavioral factors would predict higher levels of both types of SE, such as older parent age, greater social support, and more behavioral achievement during treatment (e.g., quit attempt initiated). Second, we hypothesized that greater SE to abstain from smoking would be enhanced by a combination of biological (i.e., lower baseline dependence level), behavioral (e.g., greater urge management efforts, and more days abstinent during treatment), and social factors (e.g., social support for cessation). In contrast, we hypothesized that SE to reduce children’s

Lines 119-125 describe our hypotheses in terms of measures we used in our analyses. We modified this paragraph to differentiate it from lines 55-60. It now reads: “Specifically, to guide our analyses, we hypothesized that older parent age and greater social support, as well as measures capturing short-term effort and achievement with smoking behavior change during treatment (e.g., initiating a quit attempt) would relate to both measures of self-efficacy at end-of-treatment. We hypothesized that biological (i.e., lower baseline nicotine dependence level) and social (i.e., social support for cessation) factors, in combination with the behavioral factors described above, would enhance end-of-treatment SE to abstain from smoking.”  

Detailed description and discussion is not necessary for all the factors which might affect smokers’ SE. The authors can focus on several important points.  Description regarding smoking intervention and social support in L83-90 are similar to that in L244-251. Another description regarding parents’ age in 100-111 are similar to that in L265-272.

Our intention in the discussion was to describe outcomes of our analysis. However, in this revision, we attempted to minimize repeating points about factors described in detail in the introduction. For social support, we kept most of the details in the discussion because we felt it was adding new information and explaining why that may be important to consider in an intervention. For parent age, we took out most of the detail in the discussion because it was redundant and had been previously described.